# "I'm going to stay young": Belief in anti-aging efficacy of menopausal hormone therapy drives prolonged use despite medical risks

**Mary M. Hunter** [1]*, **Alison J. Huang** [2], **Margaret I. Wallhagen** [1]

**1** Department of Physiological Nursing, School of Nursing, University of California San Francisco, San Francisco, California, United States of America, **2** Department of Medicine, School of Medicine, University of California San Francisco, San Francisco, California, United States of America

☉ These authors contributed equally to this work.
* mary.hunter@ucsf.edu

**Data Availability Statement:** All relevant data are contained within the paper. Excerpts from study transcripts constitute the minimal data set, defined

## Abstract

### Background

Over a third of menopausal hormone therapy (HT) prescriptions in the US are written for women over age 60. Use of HT more than 5 years is associated with increased risk for cardiovascular disease; breast, ovarian, and endometrial cancers; thromboembolic stroke; gallbladder disease; dementia; and incontinence.

### Objectives

To explore older women's perceptions of the benefits and risks of long-term HT and examine factors influencing their decisions to use HT > 5 years despite medical risks.

### Methods

A qualitative approach was selected to broadly explore thought processes and social phenomena underlying long-term users' decisions not to discontinue HT. Interviews were conducted with 30 women over age 60 reporting use of systemic HT more than 5 years recruited from an urban area in California and a small city in the Rocky Mountain region. Transcripts of interviews were analyzed using conventional grounded theory methods.

### Results

Women reported using HT to preserve youthful physical and mental function and prevent disease. Gynecologists had reassured participants regarding risk, about which all 30 expressed little concern. Participants, rather than providers, were the principal drivers of long-term use.

### Conclusions

Participants perceived estrogen to have anti-aging efficacy, and using HT imparted a sense of control over various aspects of aging. Maintaining this sense of control was prioritized over potential risk from prolonged use. Our findings provide an additional perspective on

as the data set used to reach the conclusions drawn in the paper.

**Funding:** MMH received a National Research Service Award from NIH/NINR 1F31NR015170-01A1. The funder played no role in study design, data collection and analysis, decision to publish, or preparation of the manuscript.

**Competing interests:** The authors have declared that no competing interests exist.

previous work suggesting the pharmaceutical industry has leveraged older women's self-esteem, vanity, and fear of aging to sell hormones through marketing practices designed to shape the beliefs of both clinicians and patients. Efforts are needed to: 1) address misconceptions among patients and providers about medically supported uses and risks of prolonged HT, and 2) examine commercial influences, such as medical ghostwriting, that may lead to distorted views of HT efficacy and risk.

## Introduction

Prolonged use of menopausal hormone therapy (HT) is common despite prescribing guidelines recommending that it be used no longer than 5 years at the lowest dose possible to relieve symptoms of menopause. [1] Prolonged HT is not recommended because use of HT more than 5 years is associated with increased risk for cardiovascular disease; breast, ovarian, and endometrial cancers; thromboembolic stroke; gallbladder disease; dementia; and urinary incontinence.[1–4]

Over a third of HT prescriptions in the US in 2015 were written for women over age 60, when the prevalence of systemic HT among women in the United States (US) with commercial health insurance ranged from 6.8 percent in women 60–64 years and 1.6 percent in women over age 74.[5] Similar prevalence data have been reported in northern European countries.[6] The US data underestimated HT prevalence due to omission of Medicare insurance data and because prescriptions for vaginal estrogen and compounded hormones were not included.

A review of the literature on HT decision making (reported elsewhere) found no studies addressing the phenomenon of long-term use, although all studies examined perception of risks and benefits.[7] Concern about increased risk for breast cancer was a dominant theme in this body of literature.[8–12] Because of this finding, we anticipated that users of prolonged HT would be concerned about their risk for breast cancer. Women in the reviewed studies expressed confusion about HT risks and benefits, citing conflicting information in the media. [8–12]

The cumulative lifetime incidence of breast cancer in the US is one in eight women.[13] Primary risk factors are age and timing of initiation and duration of HT.[14, 15] The most recent meta-analysis of worldwide evidence concluded that for lean women on HT 5 years, breast cancer incidence would increase at ages 50–69 by 1 in 50 among users of estrogen-progestin HT and 1 in 200 among users of estrogen-only HT. Excess risk at 10 years on HT would be twice as great.[15]

This paper reports findings from a qualitative study exploring reasons why women continue using HT despite risks. The study's purpose was to inform efforts to reduce prolonged HT and associated disease. To explore reasons why some women persist in using HT more than 5 years despite known health risks, two questions were addressed: "What do long-term users perceive to be the benefits and risks of HT?" and "Who and what influence women's decisions about HT?" A qualitative approach was selected to broadly explore thought processes and social phenomena underlying long-term users' decisions not to discontinue HT.

## Methods

The institutional review board (IRB) of the University of California San Francisco (UCSF) approved the study titled "Hormone Therapy Decision Making in Older Women." The IRB study number is 13–11474. The UCSF IRB also approved all associated study documents,

including the recruitment flyer and written consent form. This research report is an analysis of study data that has not previously been published.

We anticipated that a sample of 30 participants would be an adequate number for a qualitative study of HT use in older women, as participants would be drawn from a population with potentially similar demographic and personal characteristics. Demographic characteristics of the sample are presented in Table 1 at the beginning the Results section.

Thirty women were recruited from an urban area in northern California and a small city in the Rocky Mountain region, areas likely to offer a range of social and cultural perspectives. Participants responded to flyers left in public places and posted on Internet list servers including Nextdoor™ and Craigslist™. To be eligible, women had to be over age 60 or older, born female, to have taken systemic oral or transdermal HT (pills or patches) more than 5 years, to be currently using systemic HT, and to be able to read and speak English. Since there may be a unique clinical rationale for long-term HT to treat severe osteoporosis in women who cannot tolerate bisphosphonates, no women with severe osteoporosis were enrolled.

Women responding to fliers were screened for eligibility in telephone conversations, and all respondents who met inclusion criteria were invited to complete hour-long interviews with the first author. Each interview was conducted in a private space chosen by the participant following the signing of a written consent document. Each woman was asked to share what she recalled thinking and feeling as she approached menopause. A rationale for requesting this narrative was to give the participant an opportunity to guide the tone and content of the interview. This approach elicited a discussion of reasons for starting and continuing HT. The direction of each narrative was influenced somewhat by the interviewer in that prompts such as "can you tell me more about that?" were used. If a participant did not volunteer her thoughts about health risks, the interviewer asked the question, "What concerns do you have about risks associated with HT?"

Interviews were audio-recorded, transcribed verbatim, and analyzed using conventional grounded theory methods. The first author analyzed and assigned initial thematic codes to interview data after which co-authors reviewed transcript citations and provided input about themes arising from the data. Coded passages were organized using NVivo 10 software (QSR International).

**Table 1. Participant characteristics.**

| Characteristic | Number |
|---|---|
| Number of participants | 30 |
| Age | |
| Mean | 68 |
| Range | 61–80 |
| Education | |
| Some college, no degree | 2 |
| Bachelors | 11 |
| Masters | 11 |
| PhD/MD/DDS | 6 |
| Relationship status | |
| Single | 2 |
| Partnered | 21 |
| Divorced | 6 |
| Widowed | 1 |

## Results

Of 30 participants, the mean age was 68, with a range of 60–80 years. Mean duration of HT was 18 years, with a range of 5–45. Participants included a practicing internist, a nurse practitioner specializing in menopause treatment, two retired nurses, and two other retired medical professionals in addition to 24 women with no medical training or background. All but one was White, one was Asian, and all had some college education. Recruitment for a pilot study began in November 2013, and study recruitment ended in December 2017. Interviews were completed by December 2017. Demographic characteristics of participants are summarized in Table 1.

All participants voluntarily used the term "hormone replacement therapy" or "HRT" in interviews, a marketing term that implies "replacement" of a natural or necessary substance. (Although the juxtaposition of the terms HRT and HT may be confusing, this report includes both because "HRT" appears in interview transcripts, whereas HT is the accepted medical term.) The woman quoted below had experienced two mastectomies and a cholecystectomy after starting HT. She had persisted in finding a provider who was willing to prescribe estrogen despite her medical history. She said "Without HRT I would not feel feminine," and explained further:

> So it always seemed to me that estrogen was a natural substance that your body produced, and when you're young and you're healthy you feel well. When you go through menopause and you stop producing estrogen you don't feel as good, so to replace it seemed like a natural thing to me. It didn't seem foreign or harmful, because it was just giving back to you what you had before.

## Control of aging

Interview data suggested that maintaining one's sense of self was a major concern among participants. While not all credited HT for cosmetic benefit, all asserted that HT helped them maintain some aspect of physical and/or mental function they associated with youth and health. Most mentioned using HT to prevent diseases associated with aging.

**Physical activity.** Women credited HT with maintenance of athleticism, strength, and youthful energy.

> I guess if you talked to me ten years ago about that I would have said this estrogen stuff is going to keep my skin. and I'm going to stay young. . .. I'm much more adjusted to this idea of aging and being wise and not worrying. But it's more function. It's what I want to do exercise wise. It's what I want to do hiking and skiing. . .. I want to be one of these active aging people.

**Sexual health.** Participants hypothesized that if they stopped HT, they would be unable to enjoy sex due to vaginal symptoms and loss of desire.

> . . . there's vaginal dryness, which is big. You know, I still have a very nice sex life, and I'm afraid that if I stopped taking them, it would affect that. And I'm not ready to give that up. . . And so, I don't want to give that up. . .. I think if I. . . didn't have a partner that was still sexually active, I would be more willing to give it up.

Some assumed that women who did not use HT were unlikely to want sex or enjoy it, thus conveying their underlying assumption that HT was necessary to maintaining or enjoying sexual activity after menopause.

*I've only been married for three and a half years, this time, and sexual desire is a biggie, although I don't know what it would be without it, but I don't want to take a chance because of the stories that friends have told me about "Oh, I went through menopause, and I just never wanted to have sex again" and I'm like "Oh my gosh, that's really not going to work for me."*

**Skin benefits.**   Women gave HT credit for lubricating body tissues and maintaining skin elasticity. A perception that estrogen had minimized skin aging emerged in many interviews.

*I think it gives me a sense that I'm well, that that stuff is inside of me working to ease my joints, working to help my really badly sun-damaged skin. It might be lubricating my eyes. I have a feeling that estrogen is a drug that we were supposed to have if we were a woman and that yanking those parts out did mean something to my health other than periods. . .. I feel like when I put it in, I'm not harming myself. . .. My general sense is that it's helping me age easier. . .. I just have a feeling that it's lubricating me kind of from the inside out, that it kind of does that for me.*

Another woman enjoyed her youthful appearance and "vibrancy," which she attributed to HT rather than to her intrinsic health or personality.

*So I guess the reasons, if you ask me the reasons why I want to stay on HRT, I feel like it's continuing a general overall vibrancy, more youthful vibrancy than aging, and skin elasticity is one of those things. I think of myself as having more energy than my friends who are not on HRT, that I have more energy. My skin looks better than theirs.*

Women who remarked that HT benefited their skin invariably explained that this was not the primary reason they used it. They may have feared that using HT to improve their appearance belied vanity or diminished the validity of a medical indication.

*I think it certainly has helped the condition of my skin and therefore my appearance, and that's a nice side benefit. But interestingly, that wasn't really one of the big factors that caused me to decide to do HRT. It's just been sort of a, "Oh wow, that's nice" I mean, and since I had my eyes done when I was 50 and had some liposuction under my chin at that age, I can't say why I didn't think more about that, but I didn't.*

**Preservation of memory and clear thinking.**   Maintaining cognitive ability was mentioned frequently as a motivation to continue HT. Participants believed estrogen enabled them to think clearly and continue to do their jobs capably.

*I got a wonderful new Ob-Gyn who said that all of the professional women in her practice took on hormone replacement because otherwise their brains wouldn't work. I thought, "I understand, why I took it. Yes." I think there was one time when I stopped taking it, my brain turned to mush. . .. Even now that I'm retired, that's not okay for me. I do have some occasional editing, and I don't have confidence I could do it without hormone replacement.*

The woman quoted above was asked whether she believed that all women who are not using HT are unable to think clearly. "Yes," she said, "I do." The woman quoted below described a miraculous transformation in her memory, mood, and "everything" that occurred almost immediately after starting HT.

*. . . a lot of people relied on me. And I began to realize that I wasn't remembering as well as I had been able to. . . So I went to my doctor, my internist, and told her what was going on. She said she was pretty sure that I was in a perimenopause or premenopausal period. . .. Now I remember this conversation so clearly, because it was hilarious to me. She said she was going to give me a little pink pill, and that she was pretty sure that within a couple of days I would no longer have any trouble with my memory. And I thought, "Yeah, right." Okay. But I filled the prescription and began taking it. And within two days my memory was almost photographic. I mean, it was incredible, the dramatic improvement. And I had no further problem. And consequently, my mood improved dramatically. . .. I was just back. And it was fabulous. I also, of course, started sleeping really well. Everything got better.*

**Disease prevention.** Women had heard from doctors and popular media that HT may prevent cardiovascular disease, dementia, and other conditions such as hypertension and osteoporosis. Participants suggested that preventing disease balanced possible risk, suggesting a cognitive mechanism facilitating denial of risk.

*I'm not sure I'm going to be right about this, but I think I just read this in that article, that the estrogen also helps keep your blood pressure down because as we age, it's going to creep up anyway. And am I thinking correctly about that, that there's a correlation?*

The woman quoted below believed HT reduced her risk of Alzheimer's disease. Despite acknowledging cancer risk, she prioritized perceived control of Alzheimer's.

*You can threaten me, I will not stop this until I die. So, there's part of me that knows there's a risk, although I was not too much at risk because cancer in my family has not been a problem so far. . .. So I was feeling lucky in the way of I might just get away with it and hopefully survive. And at this point, if I heard that I had cancer, okay. It's the risk that I'm accepting to take, and I'd rather have cancer than Alzheimer's, because at least they let us die of cancer, and they don't let you die of Alzheimer's, so that's my choice so far. . .. I want to stay on it. I feel better with it.*

## Challenges finding prescribers

Some women had experienced health problems while using HT, and this necessitated finding new providers willing to prescribe it. Others spoke of needing to change providers to continue HT due to their age. Some had located providers out of their insurance networks willing to prescribe HT after learning their current providers would not. By switching to different providers within their current insurance networks, most participants had avoided the threat of having to pay out-of-pocket for HT. Several women used a combination of FDA-approved hormones covered by their health insurance and compounded hormones that they paid for themselves.

The woman quoted below described her network of providers, which included a gynecology specialist, internist, naturopath, and psychologist. Because her narrative suggested that she had worked hard to find providers willing to support her use of HT, she was asked whether she was directive regarding her therapy.

*I wouldn't say that (I direct my providers) so much. I will do my research and I might have a point of view about it. . . with Dr. (A), I would not propose anything different than what she's doing. She has the knowledge, she does the research, I don't. And I trust her. With Dr. (B), less*

*so. Because I don't know that she would necessarily agree with all the hormones that I take. . .. But I don't necessarily have an opinion in advance of what I want to do. I have a need, like when I had low energy and I consulted Dr. (C), but I never presume to tell—I was paying her a lot to tell me. . .. I'll change doctors until I find one that seems to agree with my point of view. But I don't recall ever directing, mainly because with Kaiser I know they're going to go by the book. They are very, as Dr. (B) says, "evidence-based.". . . I guess directive isn't a word I would use. I would say I'm more involved and concerned. And I don't automatically take what they say as truth. I'll generally do my own research on it and might consult a different doctor like Dr. (D) and Dr. (E). Probably they don't agree on the need for me to take various things. But at this point at least some research has caught up enough so that Dr. (B) believes that Climera has a—I wish I could remember that word, a way of preventing depression. Or a way of dealing—of preventing dementia. It's a protective effect against dementia. So both (A) and (B) seem of that opinion, so that's great. But if they weren't, I wouldn't argue with them. I would go to a different doctor. So I guess in that sense, I would be looking for someone that I could direct.*

### Risk—provider influence, confusion, and fear of quitting

No participant expressed concern about HT health risks without being asked, and few said they were concerned about the risk of breast cancer with prolonged use. For example, when this woman was asked whether she was concerned about increased risk of breast cancer with long-term use, she responded,

*I had forgotten that aspect of side effects, and. . . no. It's never been a concern for me. None of my immediate relatives have ever had breast cancer. It's never been an issue in my family. I remember when I was first taking the pill, I got some significant lumps in my breast because I was taking too strong a pill. But with this therapy, it's never been an issue.*

Most participants reported obtaining HT prescriptions from gynecology specialists rather than generalists/internists, and many reported that their providers had played a role in shaping positive views. The woman quoted below echoed the words of others who stated that concern about HT risk raised by results of the Women's Health Initiative (WHI) was overblown, the WHI was a flawed study, and the findings were irrelevant.

*My original doctor. . . explained the difference between the study that was done, the big quote "definitive" study, and said, "Hey, this was very different. These people didn't start taking hormones until 10 years later. . . they were on Provera and Premarin. And you've not been on either of those two, and you're on the lowest dose possible, and you seem to have no ill effects. And you have no family history and blah, blah." So I've not felt any worry about any of that.*

### Confusion resulting from conflicting data

Participants cited inconsistent information about risk as evidence that HT warnings were invalid. Risk information from providers was deemed valid as long as it was reassuring. Few participants made a distinction between the risk of short- vs. long-term use. Some recognized that they selectively accepted risk information.

*You know, over time, like 20, 30 years, there have been many conflicting reports, observations about dementia and all of the things that may be impacting it. I would be surprised if there isn't some relationship. But at this point, maybe I have chosen not to read about that because*

*I don't want to know. Having been on hormone replacement for a long time, I prefer to think there's either no impact or only a positive one on delaying dementia. But I could be wrong. As I said, I've chosen not to go there. Denial is an interesting thing, especially for a scientist.*

**Fear of quitting.** Despite acknowledging possible risks, participants expressed more concern about quitting than continuing HT. HT had given participants a sense of control that they were unwilling to give up despite health warnings and increasing pressure from provider networks. The woman quoted below admitted she was unsure whether she benefitted from HT, yet she expressed determination to continue, in part due to concerns about losing a youthful appearance if she quit.

*Maybe I'd be exactly the same now if I'd never taken a thing. And that includes vitamins, all the thousands and thousands of dollars I spent on vitamins. And hormones may have been for naught. I mean, if I had not taken anything, I might be exactly the same as I am, but I have no way to know that. . .. I look at other women my age. They look older. . .. Maybe that has nothing to do with what I take. Maybe it has everything to do. . .. I go completely on faith that I seem to be okay now, so why stop anything and risk going backwards. . .. I may not feel anything different, in which case I would say, "Well I guess that was a waste of time and money." But my intuition tells me that it's useful in some way, and so I would be upset if I were denied access to those things because I believe strongly enough that they work, and if they take them away I might be proved wrong, or I might suddenly deteriorate.*

## Discussion

This study provides new insight into the thought processes and underlying priorities of women over age 60 who opt for prolonged use of HT. Participants indicated that continuing HT gave them a sense of control over certain aspects of aging. Many had struggled to keep receiving HT prescriptions, and our data suggest such struggles were undertaken to maintain this sense of control.

Differential uptake of HT generally reflects differences between social classes, particularly in regard to the social desirability of a youthful appearance.[16] Aging impacts the self-esteem, social identity, and health of women, and our study demonstrates how fear of aging can drive health-related behavior such as prolonged HT use. Women feared discontinuing a drug purported to help them maintain youthful qualities and overall wellbeing. In many cases, long-term users prioritized the sense of control and perception of youthfulness that HT gave them over better health outcomes.

Based on our review of the literature, we anticipated that users of prolonged HT would be concerned about their risk for breast cancer, so finding that participants paid little attention to HT risk was somewhat surprising. This lack of concern about risk may reflect distrust of inconsistent scientific literature and an unwillingness to accept information that fails to reinforce a decision to initiate HT for other reasons, such as to treat vasomotor symptoms.

The literature review also suggested that women may experience confusion about HT risks and benefits due to the presence of conflicting information in the media.[8–12] Although a large body of research has addressed the risks and benefits of HT, some information on HT risks and benefits in medical literature has been shown to reflect pharmaceutical industry bias. [17–20] Medical literature that reflects industry bias and popular media that quotes this content is a potential source of confusion for users of prolonged HT.

Although our study cannot provide definitive evidence of corporate influence on prolonged use of HT, the findings are interesting in light of prior work identifying themes in industry-sponsored ghost-written literature targeted to gynecology specialists. Ghostwriting refers to the practice of paying researchers and physicians to lend their names to journal articles, abstracts for conferences, and continuing education materials that have been prepared by marketing firms hired by pharmaceutical companies. Concerns about ghostwriting in relation to HT have appeared in articles published in PLOS Medicine and The New York Times, including concerns about ties between advisory board members for gynecology specialty journals and manufacturers of HT.[17–21] Areas of focus in ghostwritten articles on HT that have relevance to our findings are 1) claims of anti-aging efficacy,[22] 2) debunking concerns about breast cancer risk,[22] and 3) HT for disease prevention.[17–20,22,23]

Other research has suggested that ghostwriting has influenced the prescribing practices of gynecology specialists.[18–23] Comments from several participants indicated that their gynecologists had played a role in shaping positive views of HT and downplaying risk. Other studies of HT prescribing practices indicate that gynecology specialists tend to prescribe HT for more women and for longer duration than general practitioners, family practice physicians, or internists.[24, 25] However, our data suggest that participants were primarily motivated by their own beliefs about HT rather than were persuaded to continue by a healthcare provider. Determined to continue HT, they had ignored information contradicting their beliefs and opted to change providers when current prescribers showed resistance.

## Strengths and limitations

This research yielded rich qualitative data from detailed interviews of 30 women. Because survey data indicate that most HT users are White and many are well-educated, our study sample was demographically representative of typical users of HT.[25] Compared to other long-term users, women who volunteered for our study may have had stronger or different beliefs about HT. It is also possible that research conducted in different geographic regions might have yielded different results.

## Conclusions

Findings from this study suggest that prolonged use of HT is driven to a large extent by societal anti-aging views and a belief in HT's anti-aging efficacy. Our findings provide an additional perspective on previous work suggesting the pharmaceutical industry has leveraged older women's self-esteem, vanity, and fear of aging to sell hormones through marketing practices designed to shape the beliefs of both clinicians and patients.[17,20] Our results also suggest that long-term users either do not believe evidence of the risks of prolonged HT or place lower priority on these risks relative to what they perceive as benefits, such as youthful appearance, energy, and identity.

Efforts are needed to clarify uses of HT that are efficacious and medically supported and to communicate the risks of prolonged HT. Because marketing messages may masquerade as science, increasing public awareness of drug marketing practices such as ghostwriting is also necessary.

## Author Contributions

**Conceptualization:** Mary M. Hunter.

**Formal analysis:** Mary M. Hunter, Alison J. Huang, Margaret I. Wallhagen.

**Funding acquisition:** Mary M. Hunter.

**Investigation:** Mary M. Hunter.

**Methodology:** Mary M. Hunter.

**Project administration:** Mary M. Hunter.

**Writing – original draft:** Mary M. Hunter.

**Writing – review & editing:** Alison J. Huang, Margaret I. Wallhagen.

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
