## [Decision Letter · Decision Letter 0]

13 Apr 2020

PONE-D-20-05741

'I’m Going to Stay Young': Belief in Anti-Aging Efficacy of Hormone Therapy Drives Prolonged Use Despite Medical Risks

PLOS ONE

Dear Dr. Hunter,

Thank you for submitting your manuscript to PLOS ONE. After careful consideration, we feel that it has merit but does not fully meet PLOS ONE’s publication criteria as it currently stands. Therefore, we invite you to submit a revised version of the manuscript that addresses the points raised during the review process.

We would appreciate receiving your revised manuscript by May 28 2020 11:59PM. To enhance the reproducibility of your results, we recommend that if applicable you deposit your laboratory protocols in protocols.io, where a protocol can be assigned its own identifier (DOI) such that it can be cited independently in the future. For instructions see: http://journals.plos.org/plosone/s/submission-guidelines#loc-laboratory-protocols

We look forward to receiving your revised manuscript.

Kind regards,

Antonio Simone Laganà, M.D., Ph.D.

Academic Editor

PLOS ONE

Journal Requirements:

2. Please provide additional details regarding participant consent. In the ethics statement in the Methods and online submission information, please ensure that you have specified what type of consent you obtained (for instance, written or verbal, and if verbal, how it was documented and witnessed).

3. In your Methods section, please provide additional information about the participant recruitment method and the demographic details of your participants. Please ensure you have provided sufficient details to replicate the analyses such as: a) the recruitment date range (month and year), b) a description of any inclusion/exclusion criteria that were applied to participant recruitment, c) a table of relevant demographic details, d) a statement as to whether your sample can be considered representative of a larger population, e) a description of how participants were recruited, and f) descriptions of the specific locations where participants were recruited and where the research took place.

4. Please provide a sample size and power calculation in the Methods, or discuss the reasons for not performing one before study initiation.

Additional Editor Comments (if provided):

The topic of the manuscript is interesting. Nevertheless, the reviewers raised several concerns: considering this point, I invite authors to perform the required major revisions.

Reviewers' comments:

Reviewer's Responses to Questions

**Comments to the Author**

1. Is the manuscript technically sound, and do the data support the conclusions?

Reviewer #1: Yes

Reviewer #2: Yes

Reviewer #3: Yes

2. Has the statistical analysis been performed appropriately and rigorously? 

Reviewer #1: I Don't Know

Reviewer #2: N/A

Reviewer #3: N/A

3. Have the authors made all data underlying the findings in their manuscript fully available?

Reviewer #1: Yes

Reviewer #2: No

Reviewer #3: Yes

4. Is the manuscript presented in an intelligible fashion and written in standard English?

Reviewer #1: Yes

Reviewer #2: No

Reviewer #3: Yes

5. Review Comments to the Author

Reviewer #1: The topic is well-introduced and the problem stated

• The methods and results are fair. The authors are kindly invited to provide the reference number and date of the ethical clearance letter

• Some of the references need revision (ref.14, and 20)

I strongly recommend the manuscript for publication

Best regards

Reviewer #2: 1) Introduction needs to revised to be more coherent and and linking to the objective of the study

2) How did authors ensure that women without severe osteoporosis did not participate in the study?

3) Were women with breast or any other female cancers included? This has to be made clear.

4) The whole manuscript has to revised fro grammatical errors.

Reviewer #3: I was pleased to revise the manuscript entitled “'I’m Going to Stay Young': Belief in Anti-Aging Efficacy of Hormone Therapy Drives Prolonged Use Despite Medical Risks” (Manuscript Number: PONE-D-20-05741).

The study was approved by the local Institutional Review Board of the University of California San Francisco. The IRB study number is 13-11474.

In general, this manuscript was aimed to explore older women’s perceptions of the benefits and risks of long-term HT and examine factors influencing their decisions to use HT > 5 years despite medical risks. In my honest opinion, the topic is interesting enough to attract the readers’ attention.

Methodology is accurate and conclusions are supported by the data analysis. Nevertheless, authors should clarify some points.

In general, the Manuscript may benefit from some revisions, as suggested below:

- Lines 9-11. I would suggest specifying “postmenopausal”.

- Methods. Based on the methods description, the ongoing intake of HT was not an inclusion or exclusion criteria of the study. I would suggest clarifying this point.

- Methods. I would suggest improving the description of the flier’s content. Did they refer to HT or HRT?

- Methods. How were the interviews conducted? Did the principal investigator follow specific questions? How did the investigators avoid guiding the affect the discussion and those reported by the women?

- Results. I would suggest, if possible, providing the number of subjects reporting each specific point. In the example, line 10, p 14, I would suggest to ad to “a few” the number and % of patients reporting this point. It is unclear the weight of each point supporting the reported opinions.

6. PLOS authors have the option to publish the peer review history of their article (what does this mean?). If published, this will include your full peer review and any attached files.

Reviewer #1: Yes: Dr. Hyder Mirghani, MD, MSc, Associate Prof. of Clinical Medicine, And Endocrine, Department of Internal Medicine, Medical College, University of Tabuk

Reviewer #2: No

Reviewer #3: No

---

## [Author Response · Author response to Decision Letter 0]

23 Apr 2020

Response to Reviewers

The authors wish to thank the Academic Editor and the reviewers for the opportunity to submit a revised manuscript. We trust that our responses clarify the important points raised in the review and improve the paper.

Points raised by Academic Editor

To enhance the reproducibility of your results, we recommend that if applicable you deposit your laboratory protocols in protocols.io, where a protocol can be assigned its own identifier (DOI) such that it can be cited independently in the future. For instructions see: http://journals.plos.org/plosone/s/submission-guidelines#loc-laboratory-protocols

PLOS guidelines for qualitative data indicate that “For studies analyzing data collected as part of qualitative research, authors should make excerpts of the transcripts relevant to the study available in an appropriate data repository, within the paper, or upon request if they cannot be shared publicly.” The authors have included relevant excerpts of transcripts within the paper (the minimal data set). Additional excerpts are included in the dissertation report referenced in the paper and accessible online (Hunter, 2018). Because full transcripts provide sensitive details and could potentially enable readers to identify individual participants and their health care providers, the authors have ethical concerns about submitting them to the Qualitative Data Repository. We have clarified this information in the online submission form by stating that "all data are fully available without restriction" (transcript excerpts). We also reported that "all relevant data are contained within the manuscript. Excerpts from study transcripts constitute the minimal data set, defined as the data set used to reach the conclusions drawn in the manuscript." 

Journal Requirements

Using the links provided, the authors have edited the manuscript in accordance with the style requirements of PLOS ONE. 

2. Please provide additional details regarding participant consent. In the ethics statement in the Methods and online submission information, please ensure that you have specified what type of consent you obtained (for instance, written or verbal, and if verbal, how it was documented and witnessed). 

Written consent was obtained using a document approved by the IRB of UCSF. We have included this information in the Methods section of the paper on page 5 lines 7-8 and page 6 lines 5-6. This information has been added to ethics statement of the online submission. 

3. In your Methods section, please provide additional information about the participant recruitment method and the demographic details of your participants. Please ensure you have provided sufficient details to replicate the analyses such as: 

a) the recruitment date range (month and year), 

Participant recruitment took place between November 2013 and December 2017. This information is on page 7 lines 4-5. 

b) a description of any inclusion/exclusion criteria that were applied to participant recruitment, 

A description of inclusion/exclusion criteria is on page 5 lines 18-21 and page 6 lines 1-2.

c) a table of relevant demographic details, 

Table 1 at the beginning of the Results section on page 7 contains relevant demographic details. 

d) a statement as to whether your sample can be considered representative of a larger population, 

A statement as to whether the sample can be considered representative of a larger population is on page 19 lines 3-7. Because our sample consisted of primarily white, well-educated women with health insurance, it was representative of the majority of women who use prolonged HT. 

e) a description of how participants were recruited, 

A description of how participants were recruited is on page 5 lines 15-21 and page 6 lines 1-5. 

f) descriptions of the specific locations where participants were recruited and where the research took place.

Descriptions of the general locations where participants were recruited and where the research took place is on page 5 lines 15-17. Although we did not include specific geographical information in the paper, we can report here that recruitment was conducted in the San Francisco Bay Area and Bozeman, Montana.

4. Please provide a sample size and power calculation in the Methods section, or discuss the reasons for not performing one before study initiation.

Sample size is addressed in the revised manuscript on page 5 lines 10-14. Journals typically accept research reports of qualitative studies that involve interviews with a sample size of 30. Qualitative scholars maintain that the concept of saturation is the most important factor in making sample size decisions. Saturation is reached ‘‘when gathering fresh data no longer sparks new theoretical insights, nor reveals new properties of your core theoretical categories’’ (Charmaz, 2006, p. 113). When a study sample is somewhat homogenous, as ours was, achieving saturation can require fewer interviews than when a sample is heterogenous. We could reasonably assert that in this study, saturation of core theoretical categories was reached at the midpoint of data collection. We found that obtaining interviews after saturation was achieved yielded additional rich quotes and strengthened theoretical findings. 

Charmaz, K. (2006). Constructing grounded theory: A practical guide through qualitative analysis. London: Sage Publications.

Review Comments to the Author

Reviewer #1 

Reviewer #1: The topic is well-introduced and the problem stated

• The methods and results are fair. The authors are kindly invited to provide the reference number and date of the ethical clearance letter

The UCSF IRB Study Number is 13-11474, and the date of the original ethical clearance letter is 8/16/2013. IRB approval was renewed several times.

The IRB Study Number is provided on page 5 lines 7. 

• Some of the references need revision (ref.14, and 20)

We are very grateful to reviewer #1 for identifying two references that should not have been in the reference list. The revised manuscript has been corrected (after being detached from Endnote), and all citations have been checked for accuracy.

I strongly recommend the manuscript for publication

Best regards

Reviewer #2 

1) Introduction needs to revised to be more coherent and and linking to the objective of the study

The introduction has been edited to better link background information to the objectives of the study. In the revised manuscript, this edited text is on page 3 lines 14-23, page 4 lines 1-22, and page 5 lines 1-2. 

2) How did authors ensure that women without severe osteoporosis did not participate in the study?

Participants were screened for eligibility in telephone conversations. No women using systemic estrogen for treatment of severe osteoporosis were included in the study sample, and no women with severe osteoporosis volunteered to be in the study. Edited text clarifying this point is on page 5 line 21 and page 6 lines 1-5.

3) Were women with breast or any other female cancers included? This has to be made clear.

A medical history of cancer was not an exclusion criterion. One participant had experienced breast cancer. See page 8 lines 2-5. 

4) The whole manuscript has to revised fro grammatical errors.

We have double-checked the manuscript to identify and correct any grammatical issues.

Reviewer #3 

I was pleased to revise the manuscript entitled “'I’m Going to Stay Young': Belief in Anti-Aging Efficacy of Hormone Therapy Drives Prolonged Use Despite Medical Risks” (Manuscript Number: PONE-D-20-05741).

The study was approved by the local Institutional Review Board of the University of California San Francisco. The IRB study number is 13-11474.

In general, this manuscript was aimed to explore older women’s perceptions of the benefits and risks of long-term HT and examine factors influencing their decisions to use HT > 5 years despite medical risks. In my honest opinion, the topic is interesting enough to attract the readers’ attention.

Methodology is accurate and conclusions are supported by the data analysis. Nevertheless, authors should clarify some points.

In general, the Manuscript may benefit from some revisions, as suggested below:

- Lines 9-11. I would suggest specifying “postmenopausal”.

“Postmenopausal” has been defined to mean the period of time after a woman has experienced 12 consecutive months without menstruation. Postmenopausal HT signifies HT used during the period of time after a woman has experienced 12 consecutive months without menstruation. (Note: this term appears only in the reference list.)

- Methods. Based on the methods description, the ongoing intake of HT was not an inclusion or exclusion criteria of the study. I would suggest clarifying this point.

We have edited the manuscript to clarify that current use of HT was part of the inclusion criteria. This text is on page 5 lines 18-21. 

- Methods. I would suggest improving the description of the flier’s content. Did they refer to HT or HRT?

Recruitment flyers stated all inclusion criteria and read as follows: “You are eligible to participate if you are age 60 or older and were born female, currently use transdermal or oral estrogen or estrogen/progestin therapy, speak and read English.” The flyer included the title of the study “Hormone Therapy Decision Making in Older Women.” It did not use the term HRT. 

- Methods. How were the interviews conducted? Did the principal investigator follow specific questions? How did the investigators avoid guiding the affect the discussion and those reported by the women?

An expanded description of how interviews were conducted is on page 6 lines 7-14. There was no list of questions. Each interview started with the principal investigator asking a participant to talk about what she was thinking and feeling as she approached menopause. One reason for requesting a narrative was to give a participant the opportunity to guide the tone and content of the interview. The direction of a narrative was influenced somewhat by the interviewer in that prompts such as “can you tell me more about that” were used. Participants were asked specifically whether they had concerns regarding HT risk if they did not mention risk. 

- Results. I would suggest, if possible, providing the number of subjects reporting each specific point. In the example, line 10, p 14, I would suggest to ad to “a few” the number and % of patients reporting this point. It is unclear the weight of each point supporting the reported opinions.

We appreciate this reviewer's comment; however, we respectfully suggest that the approach described is more suited to quantitative than qualitative analysis. Our conclusions are not based on a mathematical weighting of the number of participants who expressed a theme.

---

## [Decision Letter · Decision Letter 1]

12 May 2020

"I’m going to stay young":  Belief in anti-aging efficacy of menopausal hormone therapy drives prolonged use despite medical risks

PONE-D-20-05741R1

Dear Dr. Hunter,

We are pleased to inform you that your manuscript has been judged scientifically suitable for publication and will be formally accepted for publication once it complies with all outstanding technical requirements.

With kind regards,

Antonio Simone Laganà, M.D., Ph.D.

Academic Editor

PLOS ONE

Additional Editor Comments (optional):

Authors performed the required corrections, which were positively evaluated by the reviewers. I am pleased to accept this paper for publication.

Reviewers' comments:

Reviewer's Responses to Questions

**Comments to the Author**

1. If the authors have adequately addressed your comments raised in a previous round of review and you feel that this manuscript is now acceptable for publication, you may indicate that here to bypass the “Comments to the Author” section, enter your conflict of interest statement in the “Confidential to Editor” section, and submit your "Accept" recommendation.

Reviewer #1: All comments have been addressed

Reviewer #2: All comments have been addressed

Reviewer #3: All comments have been addressed

2. Is the manuscript technically sound, and do the data support the conclusions?

Reviewer #1: Yes

Reviewer #2: Yes

Reviewer #3: Yes

3. Has the statistical analysis been performed appropriately and rigorously? 

Reviewer #1: Yes

Reviewer #2: N/A

Reviewer #3: Yes

4. Have the authors made all data underlying the findings in their manuscript fully available?

Reviewer #1: Yes

Reviewer #2: Yes

Reviewer #3: Yes

5. Is the manuscript presented in an intelligible fashion and written in standard English?

Reviewer #1: Yes

Reviewer #2: Yes

Reviewer #3: Yes

6. Review Comments to the Author

Reviewer #1: Comments were addressed, the authors provide the ethical details requested and the references were corrected

Reviewer #2: (No Response)

Reviewer #3: I was pleased to revise the manuscript entitled “'I’m Going to Stay Young': Belief in Anti-Aging Efficacy of Hormone Therapy Drives Prolonged Use Despite Medical Risks” (Manuscript Number: PONE-D-20-05741).

The study was approved by the local Institutional Review Board of the University of California San Francisco. The IRB study number is 13-11474.

In general, this manuscript was aimed to explore older women’s perceptions of the benefits and risks of long-term HT and examine factors influencing their decisions to use HT > 5 years despite medical risks. In my honest opinion, the topic is interesting enough to attract the readers’ attention.

Methodology is accurate and conclusions are supported by the data analysis. Moreover, I appreciated the manuscript improvement and answers of authors.

7. PLOS authors have the option to publish the peer review history of their article (what does this mean?). If published, this will include your full peer review and any attached files.

Reviewer #1: Yes: Hyder O Mirghani, Associate Prof. of Medicine and Endocrine, University of Tabuk, Saudi Arabia

Reviewer #2: No

Reviewer #3: No

---

## [Editor Report · Acceptance letter]

15 May 2020

PONE-D-20-05741R1 

"I’m going to stay young":  Belief in anti-aging efficacy of menopausal hormone therapy drives prolonged use despite medical risks 

Dear Dr. Hunter:

I am pleased to inform you that your manuscript has been deemed suitable for publication in PLOS ONE. Congratulations! Your manuscript is now with our production department. 

With kind regards,

on behalf of

Dr. Antonio Simone Laganà 

Academic Editor

PLOS ONE